# *In vitro* and *in vivo* characterization of wild type BMP9 and a non-osteogenic variant in models of pulmonary arterial hypertension

Tobias G. Schips ㉫*, Karl W. Kavalkovich, Lai-ming Yung, Salam Ibrahim, Makhosi Edmondson, Zhigang Hong, Chin-hu Huang, Simon Hinke ㉫, Xinkang Wang, Andrea R. Nawrocki, Annmarie Winkis, Jinquan Luo, Iman Farasat, Brian Geist, Yang Wang, Russell Bialecki ㉫, Jey R. Jeyaseelan, David R. Bauman

Janssen Research & Development, LLC, Spring House, Pennsylvania, United States of America

* tschips@its.jnj.com

## Abstract

Endothelial dysfunction and the resulting vascular remodeling are hallmarks of pulmonary hypertension, a debilitating disease of high arterial pressure in the lungs and the right side of the heart. Mutations in the BMPR2 signaling pathway are associated with the development of pulmonary arterial hypertension. Previous pre-clinical studies demonstrated that exogenous administration of recombinant human wild type BMP9 (WT BMP9) enhances BMPR2/ALK1 mediated signaling and reverses experimental pulmonary hypertension in rat models. However, BMP9 induces osteogenic activity in progenitor cells through activation of ActR2A and ActR2B receptor complexes potentially leading to unwanted bone formation in non-osteogenic tissues.

The cellular activity of human WT BMP9 and amino acid substitution variants was characterized *in vitro* in terms of BMPR2 and ActR2 signaling. We identified a mutant variant of human BMP9 that maintains its activity in endothelial cells, specifically preserving BMPR2 signaling while eliminating osteogenic signaling associated with ActR2A/B activation in mesenchymal precursor cells.

Rat models of pulmonary hypertension served as *in vivo* models to characterize efficacy and safety of BMP9 supplementation therapy. While WT BMP9 effectively activates BMPR2 signaling across species in rat, cynomolgus monkey and human systems, our human BMP9 mutant variant is inactive on rat BMPR2/ALK1 receptor complexes. Therefore, WT BMP9 was used to examine disease reversal in the preclinical monocrotaline model rat of pulmonary hypertension. WT BMP9 failed to improve right ventricular systolic pressure or right ventricular hypertrophy, despite clear target engagement shown by upregulation of SMAD7.

**Data availability statement:** All relevant data are within the manuscript and its Supporting Information files.

**Funding:** All authors are or were employees of Johnson and Johnson Pharmaceutical Research and Development: Janssen Research and Development LLC. This study was funded by Johnson and Johnson Pharmaceutical Research and Development: Janssen Research and Development LLC. The funders had no role in study design, data collection and analysis, decision to publish, or preparation of the manuscript.

**Competing interests:** All authors are or were employees of Johnson and Johnson Pharmaceutical Research and Development: Janssen Research and Development LLC. This study was funded by Johnson and Johnson Pharmaceutical Research and Development: Janssen Research and Development LLC. The funder provided support in the form of salaries for authors, but did not have any additional role in the study design, data collection and analysis, decision to publish, or preparation of the manuscript. The specific roles of these authors are articulated in the 'author contributions' section. This does not alter our adherence to PLOS ONE policies on sharing data and materials.

Telemetry studies of WT BMP9 in the Sugen 5416 and hypoxia rat model of pulmonary hypertension indicated no significant change in pulmonary pressure but led to increased systemic blood pressure and decreased heart rate. Additionally, escalating doses in naive rats caused severe dose-limiting effects and morbidity at 500 µg/kg/day or higher.

Given these findings including the absence of therapeutic efficacy in a relevant PAH animal model and dose limiting toxicity in rats, a therapeutic window for BMP9 treatment could not be established.

## Introduction

Pulmonary arterial hypertension (PAH) is a progressive and life-threatening disorder characterized by elevated blood pressure (BP) within the pulmonary arteries leading to right ventricular failure and death. The prevalence of PAH ranges from 0.4 to 1.4 per 100,000 persons [1]. Genetic and environmental factors play a role in the induction of endothelial dysfunction, vascular remodeling and inflammation, leading to vasoconstriction, neointima formation, medial hypertrophy and adventitial expansion. This vascular remodeling drives the increase in pulmonary vascular resistance, which ultimately leads to right ventricular hypertrophy (RVH) and right heart failure if left untreated. Current standard of care treatments focus on modulation of vascular hemodynamics through three main pathways: endothelin-, nitric oxide- and prostacyclin [2]. Despite significantly improving outcomes in PAH patients, current therapies, including combination therapies, are not curative and the disease has persistent high morbidity and mortality rates [3]. PAH remains a progressive disease. Two pathways have emerged as drivers of the progressive vascular remodeling observed in PAH: the platelet-derived growth factor receptor (PDGFR) and the transforming growth factor β (TGFβ) superfamily. The PDGFR pathway is overactivated and inhibition with imatinib, a small molecule inhibitor targeting multiple tyrosine kinases, in the IMPRES trial resulted in improved exercise capacity and hemodynamics in PAH patients, but serious adverse events were common [4]. Upregulation of TGFβ signaling in the pulmonary vasculature leads to endothelial dysfunction, vascular smooth muscle cell proliferation and increased deposition of extracellular matrix proteins [5], while downregulation of bone morphogenic protein (BMP) signaling, which generally exerts protective roles by inhibiting excessive cell growth and fostering vasodilation [6, 7]. Lung tissue samples from PAH patients and animal models of experimental PAH demonstrate increased expression of TGFβ family ligands, receptors and downstream mediators [5]. The experimental enhancement of TGFβ/Activin signaling in animal models exacerbates pulmonary vascular remodeling mimicking PAH pathology [8] and reduction of TGFβ signaling through genetic or pharmacological intervention was efficacious in reversing the vascular remodeling in PAH animal models [5, 9]. Recently, sotatercept, an activin ligand trap capable of reducing TGFβ/activin signaling, improved exercise capacity in PAH patients and was recently approved for the treatment of PAH [8].

BMP and TGFβ ligands are known for their promiscuous nature, as they can bind to and activate various TGFβ super-family receptors, including different types of type 1- and type 2 receptors, as well as various co-receptors leading to a spectrum of signaling responses that influence cellular processes such as proliferation, differentiation, and apoptosis [10]. The bone morphogenic protein receptor 2 (BMPR2) signaling pathway is a crucial regulator of pulmonary vascular homeo-stasis by regulating endothelial cell function and survival, vascular smooth muscle cell proliferation and extracellular matrix remodeling [11]. Loss of BMPR2 signaling disrupts vascular homeostasis by inducing endothelial dysfunction and vascular smooth muscle proliferation, leading to increased vascular resistance and contributing to the development of PAH. Fur-thermore, the importance of BMPR2 in the pathological development of PAH is highlighted by genetic mutations in sig-naling components of the BMPR2 signaling pathway, which are prevalent in heritable PAH patients [6, 7]. Loss of function mutations in BMPR2, a receptor through which BMP9 and BMP10 modulate endothelial cell signaling [12], are the most commonly associated genetic mutations with the development of PAH [7]. Additionally, mutations within the BMPR2 path-way associated with PAH include the type I receptor, activin receptor-like kinase 1 (ALK1), type III receptor accessory pro-tein (endoglin), small mother against decapentaplegic (SMAD) and BMP9 [13]. Pre-clinical work demonstrated that rescue of BMPR2 signaling or enhancement of endothelial BMPR2 activity with BMPR2 replacement or BMP9 supplementation halted and reversed PAH disease pathogenesis [14, 15]. BMP9 belongs to the TGFβ superfamily and acts selectively on vascular endothelial cells, inhibiting apoptosis, migration, and angiogenesis through activation of BMPR2 [16].

Activation of BMP9/BMPR2 downstream signaling is mediated through SMAD transcription factors (SMADs 1, 5, or 8). BMP9/BMPR2/ALK1 signaling is predominantly mediated through the transcriptional activity of SMAD1/5/8 in endothelial cells [14, 17, 12]. In addition to the normal function of endothelium protection which is mediated through BMP9/BMPR2/ALK1 com-plexes, BMP9 activates ALK1 receptor complexes with activin type 2 receptor (ActR2) A and ActR2B in osteogenic progenitor cell types to promote bone formation [18]. Systemic delivery of BMP9 to enhance or induce endothelial BMPR2 as a potential therapeutic strategy for vascular diseases like PAH could lead to unwanted bone formation. Thus, BMP9 mutant polypeptides that maintain endothelial cell signaling activity (BMPR2) but are devoid of osteogenic activity (i.e., ActR2A/ActR2B) would be required for the treatment of vascular diseases to minimize the chance of bone formation in non-osteogenic tissues.

We describe the discovery of mutant BMP9 dimers with preserved endothelial cell signaling activity through phosphory-lation of SMAD1/5/8 (pSMAD) and reduced or eliminated osteogenic activity in normal human bone marrow derived mes-enchymal stem cells in vitro. Furthermore, we demonstrated that targeting endothelial dysfunction by activation of BMPR2 with WT BMP9 was ineffective in reversing or preventing the development of PAH in rat models. Unexpectedly, significant adverse reactions were observed as a consequence of systemic administration of WT BMP9 in rats further limiting the therapeutic potential of BMP9 therapy.

## Materials and methods

### Human/Rat phospho-SMAD1/3 AlphaLISA

The AlphaLISA SureFire Ultra –pSMAD1 (Ser463/465; Revvity) and pSMAD3 (Ser423/425; Revvity) were used for evalu-ation of cell-based pSMAD1/3 activity alongside recombinant human BMP9 (R&D Systems, WT BMP9 reconstituted at 10 µg/mL in sterile 1x PBS + 4mM HCl + 0.1% BSA) as the positive control (final concentration: 10ng/mL) and vehicle (sterile 1x PBS + 4mM HCl + 0.1% BSA) as the negative control. HULEC-5a human lung endothelial cells (ATCC) or rat primary lung microvascular endothelial cells (Cell Biologics) were plated in a 96-well plate (Flat bottom, TC treated, Corning) at a density of 50,000 cells per well in 200uL growth medium (HULEC: MCDB131 Media (Gibco), + 10mM L-Glutamine (Gibco), 1% Pen Strep (Gibco) 1 µg/mL hydrocortisone (Stem Cell Technologies), 10ng/mL EGF (Sigma), 10% FBS (Gibco); or for rat cells: complete rat endothelial cell medium with kit, Cell Biologics). Cells were incubated overnight at 37°C in 5% $CO_2$. Test compounds were diluted in 100uL medium per well (MCDB131 media + 0.1% BSA (bovine serum albumin fraction V (7.5% solution) Gibco). Culture medium was removed from the cells and 100uL stimuli-medium mix was added to each

well and incubated for 1h at 37°C in 5% $CO_2$. Cells were washed once with PBS and 50 μL lysis buffer was added to each well (AlphaLISA Lysis Buffer). Plates were sealed and incubated at with shaking for 10 min at room temperature (RT). Samples were homogenized by pipetting up and down (3 times) and cell lysates were transferred (10 μL of cell lysate) to 384-well plate (Revvity) for detection of AlphaLISA signal. 5uL acceptor mix (reaction buffer 1: 2.35 μL reaction buffer 2: 2.35 μL; activation buffer: 0.2 μL; acceptor beads: 0.1 μL) were added to each well. Plates were sealed and incubated at RT on a plate shaker (350 rpm) for 1h. With reduced lighting, 5uL donor mix (dilution buffer: 4.9uL; donor beads: 0.1 μL) were added to each well; plates were sealed with aluminum foil and incubated at RT overnight on a plate shaker (350 rpm). An alpha-enabled plate reader (Envision 2105 multilabel reader, Revvity) was used to measure raw fluorescent units (RFU) with Alpha technology settings. The background control value was calculated from the RFUs of wells with vehicle only. The EC50 for each agonist was calculated using nonlinear regression and is expressed as the nanomolar concentration of the compound.

### *In vitro* gene expression assay

Human pulmonary artery endothelial cells, hPAEC (Cell Applications) were treated with WT BMP9 (R&D Systems) or BMP9-A347E at a single dose of 10nM for 24h. Following treatment, total RNA was isolated using QIAzol Lysis Reagent (Qiagen) and RNeasy Mini QIAcube Kit (Qiagen). The cDNA was prepared using SuperScript™ IV VILO™ Master Mix (Thermo). Gene expression was performed by RT-PCR using QuantStudio 12K Flex (Applied Biosystems) and Taqman probes (Hs01060665_g1 for ACTB, Hs04187239_m1 for ID2, Hs00932747_m1 for TGFBI, Hs00167155_m1 for PAI-1). The relative expression levels were determined by normalization to the expression of a housekeeping gene (ACTB) and calculation of ddCt values.

### Human osteogenesis assay

Frozen normal Human Bone Marrow Derived Mesenchymal Stem Cells (hMSC; Lonza) were differentiated towards osteogenic linage. Cells were maintained in growth media (DMEM (Invitrogen), 10% HI FBS (Gibco), 1% P/S (Invitrogen) and differentiated using osteogenic media (MSC BM Basal media (Lonza), hMSC Osteogenic Bullet kit (Lonza) following the manufacturer instructions. Compound stimulation and induction of osteogenic differentiation were initiated and 48h later, cells were lysed using the Protein Quant Sample Lysis Kit (PLA; ABI). PLA lysates were used for cDNA generation using SuperScript™ IV VILO™ Master Mix (Thermo). Gene expression was performed by RT-PCR using QuantStudio 12K Flex (Applied Biosystems) and using TaqMan PCR primer probes for a relevant osteogenic marker SP7 (osterix; Hs01866874_s1), and housekeeper gene GAPDH (Hs9999905_m1). Reported outcome measurements were gene expression levels of gene targets normalized to housekeeper GAPDH expressed as Relative Fold Change (ddCt) to vehicle negative control.

### Animal welfare statement

All studies were performed in compliance with the Animal Welfare Act regulations (Code of Federal Regulations, Title 9). All rodent studies were approved by the Institutional Animal Care and Use Committee at Janssen Pharmaceutical R&D (Spring House, PA). Studies in cynomolgus non-human primates were conducted at Charles River Laboratories under the Testing Facilities IACUC and after approval by the Janssen Pharmaceutical Animal Care & Use Council (PACC).

### Rat safety pharmacology study

This rat safety study was performed at Charles River Laboratories, planned as a 5-day exploratory toxicity study followed by a 2-week observation period. 30 female SD rats were dosed via continuous IV infusion via a femoral catheter. Animals were individually housed in solid bottom cages with certified bedding material. Individual housing was necessary due to the equipment used to facilitate tethered dose administration.

This study was performed in rats as the WT human BMP9 protein is cross reactive in rats. This study evaluated any potential toxicity associated with the target after saturating it (hence continuous infusion) at the three dose levels evaluated in this study. Previously this compound had been evaluated in pharmacology/efficacy studies up to a dose level of 30 µg/kg administered once daily subcutaneously over 21 days and was not associated with any adverse findings. This study helped in de-risking the target and identifying potential liabilities that maybe associated with BMP9 activation. The current state of scientific knowledge and the applicable guidelines cited did not provide acceptable alternatives, in vitro or otherwise, to the use of live animals to accomplish the purpose of this study.

All animals were observed at least twice a day for morbidity, mortality, injury, and availability of food and water. Veterinary care was available throughout the course of the study, and animals were examined by the veterinary staff as warranted by clinical signs or other changes. In the event that animals show signs of illness or distress, the responsible veterinarian may make initial recommendations about treatment of the animal(s) and/or alteration of study procedures, which must be approved by the Study Director. Animals were subject to Testing Facility SOP criteria and procedures for early euthanasia or when found dead. A veterinary consultation was not required if the samples were collected following anesthesia or euthanasia. Euthanasia was done by carbon dioxide inhalation followed by a Testing Facility SOP approved method to ensure death, e.g., exsanguination. If an animal is determined to be in overt pain/distress or appears moribund and is beyond the point where recovery appears reasonable, the animal will be euthanized for humane reasons in accordance with the American Veterinary Medical Association (AVMA) Guidelines on

### Euthanasia

Mortality was noted at 500 and 1500 µg/kg/day. For 500 mg/kg/day, 5 animals were euthanized in extremis on Day 4 while 4 animals were found dead on Day 3 or 4. For 1500 µg/kg/day, 2 animals were euthanized in extremis on Day 4 and 3 animals were found dead on Days 3 and/or 4. Therefore, dosing was discontinued on Day 4 for animals in all groups, including vehicle controls. Due to mortality and clinical observations indicative of overt toxicity, all surviving animals were necropsied on Day 4.

This study was approved by the Institutional Animal Care and Use Committee at Janssen Pharmaceutical R&D (Spring House, PA). All research staff involved in *in vivo* experiments received special training in animal care and handling.

### Monocrotaline (MCT)-rat model and PAH phenotype assessment

Male Sprague-Dawley (SD) Rats (180-200g) were ordered from Charles River Laboratories. Rats were allowed to acclimate to the new environmental conditions of 12h light-dark cycle at 18–20°C and 40–50% humidity for one week prior to the study and maintained or regular rodent chow (5K75 LabDiet). Animals were randomized based on body weight and injected subcutaneously (SC) with 60 mg/kg MCT (Oakwood Chemical; in PBS, pH 7.4, prepared fresh) or vehicle. On the same day, MCT rats were randomized to receive daily treatment of vehicle (50mM sodium acetate, 5% dextrose and 0.1% rat serum albumin (RSA) at ~pH 5, SC, once daily), WT BMP9 (1, 3, 10, 30 µg/kg, SC, once daily) or Imatinib (methane sulfonate salt given at 100 mg/kg, orally, once daily) for 21 days. Rats were monitored under standard animal housing conditions for 3 weeks and body weight and health status was checked twice a week. On Day 21 after MCT administration, rats were anesthetized using isoflurane (5%) and transferred to the surgical table equipped with a 37°C heating pad and a nose cone to maintain an anesthetic plane at ~3% isoflurane. Eye ointment was applied prior to the surgery. Hemodynamic measurements were performed via a non-open chest procedure. Once the animal was fully anesthetized, the abdominal cavity was surgically opened, and the sternum was kept in place using a retractor. RV apex was stabilized with a 6.0 suture and anchored with tape onto the surgical platform. An apical stab was performed using a 25G 3/4" needle. A Millar catheter was inserted gently into the stab. Right ventricular systolic (RVSP) and heart rate HR were recorded for 5–10 min to ensure stable tracing. At the end of the hemodynamic study, rats were euthanized, and the heart was dissected for RVH assessment. The weights of right ventricles

were measured for Fulton index (weight ratio of right ventricular versus left ventricular plus septum). Heart weights were also normalized to body weight and compared across treatment groups.

## Hemodynamic assessments in the Sugen SU5614/Hypoxia (Su/Hx) PAH rat model

Male SD rats (Charles River Laboratories) around 250 grams were implanted with dual telemetry devices 9HD-S21 (Data Science International) at Envigo and shipped to our institution after at least 10 days of recovery. Animals were single housed and allowed to acclimate for 1 week to standard room conditions and regular chow diet (5K75 LabDiet). After confirmation of the dual telemetry implantation by baseline BP recordings and analysis (Data Matrix 2.0, Ponemah P3; Data Science International), the telemetry-instrumented rats were injected SC with 20 mg/kg SU5614 (ApexBio) dissolved in DMSO and brought to 20 mg/ml in PBS) and placed in hypoxia chambers at 10% $O_2$ for 3 weeks. Environment factors ($O_2$, $CO_2$, humidity and temperature) in the hypoxia cages, and the body weights and health status were recorded weekly. After 3 weeks of maintenance at in hypoxic conditions, rats were returned to normoxia for 2 weeks, allowing further development of the PAH phenotype before initiating compound treatment. Rats (n = 4/group) were injected with a single dose or repeat dose of vehicle (50mM sodium acetate, 5% dextrose and 0.1% RSA at ~pH 5) or WT BMP9 (3, 10, 30 and 100 μg/kg, SC)

To assess changes in systemic hemodynamics. The baseline systemic blood pressure (SBP) and heart rate (HR) were recorded over 6h prior to dosing and 24 or 48h after each dose in repeat dosing studies. Data was summarized as 60 min averages change vs baseline. Each animal was normalized to its own baseline first, then the data of the whole group for each treatment response was compared to its own vehicle.

## Cynomolgus monkey (cyno) Pharmacokinetics (PK)/Pharmacodynamics (PD) study

This study was designed to use the fewest number of animals possible, consistent with the objective of the study, the scientific needs, and contemporary scientific standards. Although the beagle is the usual non-rodent model used for evaluating the toxicity of various test articles and for which there is a large historical database, the monkey was selected specifically for use in this study because the test agent is only reactive to NHP or humans (none of lower species), therefore, cyno has been selected. Environmental enrichment and enrichment foods were provided during the study (cognitive, social, edible, structural or manipulanda). This specific study proposal has been reviewed and approved by the NHP Study Committee at Janssen Pharmaceuticals.

Eighteen male, non-naïve cynomolgus non-human primates were randomly assigned to this PK/PD study performed by Charles River Laboratories. The animals weighed between approximately 2–6 kg. All monkeys were housed individually in stainless steel cages with environmental enrichment provided during the study. The animals were housed in a temperature and humidity-controlled room with 12h/12h light/dark schedule per day. Food and water were available ad libitum (Lab Diet Certified Primate Diet #5048; PMI Nutrition International) and provided to all animals twice daily. All animals were observed at least twice a day for morbidity, mortality, injury, and availability of food and water. Any animals in poor health were identified for further monitoring, treatment, and/or possible euthanasia. If an animal was determined to be in overt pain/distress or appears moribund and was beyond the point where recovery appeared reasonable, the animal was euthanized for humane reasons in accordance with the American Veterinary Medical Association (AVMA) Guidelines on Euthanasia and with the procedures outlined in the protocol. Animals were dosed SC or intravenous (IV) with WT BMP9, BMP9 variant or vehicle (50:50 PEG400/ Hydroxypropyl-Beta-Cyclodextrin 5% in 10mM pH 7 phosphate buffer). Animals were euthanized under ketamine anesthesia and tissue samples were collected 6h post dose. The cranial and the right lobe of the lung were preserved in RNA later at 2–8°C and frozen the next day at −80°C.

## *In vivo* gene expression analysis

The left rat lung or the cranial lobe of the left cyno lung (200−300 mg sample) were harvested, snap frozen in liquid nitrogen and stored at −80°C. While keeping tissues frozen on dry ice, a small (~3mm$^2$) sample of lung tissue was excised

from each total lung sample and placed in a pre-labeled Lysing matrix D tubes (MP-Bio) with 350uL RLT plus buffer (RNeasy Plus Mini kit; Qiagen) with β-mercaptoethanol. A FastPrep-24 5G instrument (MP Bio) was used to lyse and homogenize the tissue samples. The recommended program settings for lung tissue were used (Speed 6.0m/sec, Adapter Quickprep, Time 40 sec). Samples were transferred to Qiagen Qiashredder columns (Qiagen) and centrifuged in a bench-top microcentrifuge as per recommended settings (max speed for 2 min) to further homogenize samples. Supernatants were transferred to gDNA Eliminator mini-columns and continued through RNeasy Plus Mini kit RNA isolation steps (Qiagen) according to the manufacturer's instructions. RNA was eluted from spin columns with 50uL/tube of RNAase free H2O and quantified. RNA samples (2 µg) were used for transcription to cDNA using the Vilo Reverse-Transcription Protocol (SuperScript™ VILO™ MasterMix; Invitrogen) according to the manufacturers' instructions.

Gene expression analysis was performed using TaqMan primer-probe combinations. The PCR reactions were carried out according to the manufacturers' instructions in 384-Well Real-Time PCR Plates (Applied Biosystems), using 2X TaqMan Universal PCR Master Mix (Applied Biosystems) in a ViiA7 PCR Machine (Applied Biosystems). Taqman probes (rat: GAPDH, Rn1775763_g1; PPIL2, Rn01404147_m1; BMPR2, Rn01437215_m1; HEY1, Rn00468865_m1; Smad7, Rn01523958_m1; PAI-1, Rn01481341_m1; CHRDL2, Rn01510694_m1; FN1, Rn00569575_m1; cyno: ACTB, Mf04354341_g1; Smad7, Mf00998193_m1) were used for PCR reactions. The relative expression levels were determined by normalization to the expression of a housekeeping gene (GAPDH/PPIL or ACTB) and calculation of ddCt values.

## BMP9 production in *Escherichia coli* (*E. coli*)

Codon-optimized BMP9-A347E was cloned in topET30b vector and transformed in BL21 cells for production in *E. coli*. Inclusion bodies were isolated in lysis buffer (50mM Tris, 150mM NaCl, pH8.0) and lysed in a high-pressure homogenizer (ATS), followed by centrifugation and re-suspension in washing buffer (50mM Tris-HCl, 150mM NaCl, 10mM EDTA, 1% Triton X-100). Inclusion bodies were collected and stored in storage buffer (50mM Tris, 150mM NaCl, pH 8.0). Inclusion bodies were denaturized (20mM Tris, 7M Gua-HCl, 10mM DTT, pH8.0) followed by refolding in (25mM Tris, pH8.3, 4% CHAPS, 2mM EDTA, 1M NaCl, 10% Glycerol) at 4°C for 9 days. Protein was purified via ion exchange column (HiTrap SP FF; Cytiva) followed by size exclusion purification via Superdex75 column (Cytiva). Protein purity was confirmed to be > 90% by SDS-PAGE and correct mass was confirmed by LC-MS and SEC-HPLC. Endotoxin levels were below 1EU/mg and proteins were stored in PBS, 4% Sucrose, 0.1mM EDTA at pH7.4.

## Plasma and tissue PK measurements

Lung tissue samples (right rat- or cyno lungs) were cut into pieces and 50 mg of tissue was placed into lysing matrix D tubes (MPBio) and 500 µL of lysis buffer ((50mM Tris, 150mM NaCl, pH7.4) supplemented with Protease Inhibitor mini tablets; Pierce). Tubes were placed into a FastPrep-24 5G machine (MP Bio) and spun at speed-setting 4 for 30sec and repeated twice. Samples were centrifugated at 15000 rpm for 15 min at 4°C. Protein concentrations were measured using a BCA assay (Pierce) according to the manufacturer instructions. Individual lysate samples were diluted to 15 mg/mL.

To determine WT BMP9 concentrations in lung tissue lysate- and plasma samples, 96-well Streptavidin Gold Plates (MSD) were blocked with 150 µL/well of assay buffer (1x PBS (Gibco) + 1% BSA (Sigma) + 0.5% Tween-20 (Sigma)) and incubated for 30 min at RT with shaking at 650 rpm. Assay buffer was removed and 50 µL/well of capture reagent (Biotin-anti-hBMP9; R&D Systems – specific for human BMP9 allowing for differentiation of human, rat and cyno proteins) was added at 1ug/mL in assay buffer and incubated at RT for 45 min with shaking at 650 rpm. Plates were washed 3x with 300uL/well 1xPBS + 0.05% Tween-20 wash buffer. A WT BMP9 standard curve was prepared in 20% rat plasma (BioIVT) diluted in low cross buffer (Boca). Study samples were diluted (30 µL sample + 120 µL assay buffer). 50 µL/well standards and study samples were added to the MSD plates and incubated for 60 min at RT with shaking at 650 rpm. Plates were washed 3x with 300 µL/well 1x PBS + 0.05% Tween-20 wash buffer. 50 µL/well of detection reagent (0.25 µg/mL; Sulfo-Tag anti-human BMP9 (3209-Janssen)) was added and incubated at RT for 45 min with shaking at 650 rpm. Plates were

washed 3x with 300 µL/well 1x PBS + 0.05% Tween-20 wash buffer. 150 µL/well of 1x read buffer (MSD) was added and plates were read on an MSD Sector Imager.

## Statistical analysis

GraphPad Prism was used for statistical analyses. The data was presented as mean ± standard derivation. The difference between multiple groups was compared using One-way ANOVA or Two-way ANOVA followed by Tukey's multiple comparison test, respectively. A significance level of $p < 0.05$ was considered statistically significant.

## Results

### Pharmacological characterization of WT BMP9 and A347E variant *in vitro* and *in vivo*

The crystal structure of the BMP9/ActR2B/ALK1 tertiary complex was compared to the BMPR2 structure and the conserved ligand binding domain was used to identify the core epitope that mediates critical contacts between BMP9/BMPR2 as compared to BMP9/ActR2B [19, 20].

An amino acid substitution screen was performed by introducing point mutations (each position mutated to all other 18 amino acids except Cysteine resulting in 518 mutant variants) in the BMPR2/ALK1 binding sites of BMP9, to identify variants with reduced osteogenic properties and preserved activation of SMAD1 in endothelial cells [20]. All experiments were performed using human BMP9 proteins (WT and A347E) and differential potencies in human and rat cellular systems were observed (Table 1).

The A347E variant of human BMP9 (henceforth: A347E variant) was identified as a lead molecule based on biochemical and biophysical properties [20]. To evaluate the osteogenic activity of WT BMP9 and the A347E variant in a human cell context, mesenchymal stem cells were differentiated towards osteogenic linage and SP7 transcription factor mRNA was measured as a marker of differentiation [21]. WT BMP9 potently activated SP7 transcripts, whereas the A347E variant did not show any activity in this assay (Fig 1A), confirming the loss of osteogenic potential in a relevant human cell-based system. Furthermore, an alkaline phosphatase activity assay confirmed the findings [20].

The potency of A347E variant was compared to WT BMP9 in the HULEC-5a endothelial cell line for phosphorylation of SMAD1 and SMAD3 (Fig 1B and S1A Fig). WT BMP9 and A347E variant showed similar activities in both assays based on EC50 potency values, confirming preserved activation of the SMAD1- and 3 pathways in human endothelial cells. In addition, both WT BMP9 and A347E variant increased transcriptional activation of ID2, TGFBI and PAI-1 in human primary pulmonary endothelial cells *in vitro* (Fig 1C-E), confirming functional activation of BMPR2 downstream transcriptional target genes.

In rat endothelial cells, WT BMP9 potently activated pSMAD1; howeverA347E variant was inactive in this assay (Fig 1F), suggesting preserved SMAD1 activity in human cellular systems but loss of potency in rodent endothelial cells (summarized in Table 1) and confirmed *in vivo* in rats. WT BMP9 (30 µg/kg) and A347E variant (30 µg/kg) were administered SC to rats and lung tissue was evaluated for BMPR2 pathway activation by measuring SMAD7 transcriptional changes 6 hours post dosing. WT BMP9 showed 2-fold transcriptional activation of SMAD7; however, the A347E variant was inactive *in vivo* (Fig 1G), confirming the lack of functional BMPR2 pathway activation in rats with A347E variant.

**Table 1. Summary of cell-based assay EC50 results for WT BMP9 and A347E variant.**

| Assay | WT BMP9 [nM] | A347E BMP9 [nM] |
|---|---|---|
| Human SP7 | 0.32 | N/A |
| Human pSMAD1 | 3.7 | 5.7 |
| Rat pSMAD1 | 204.3 | N/A |

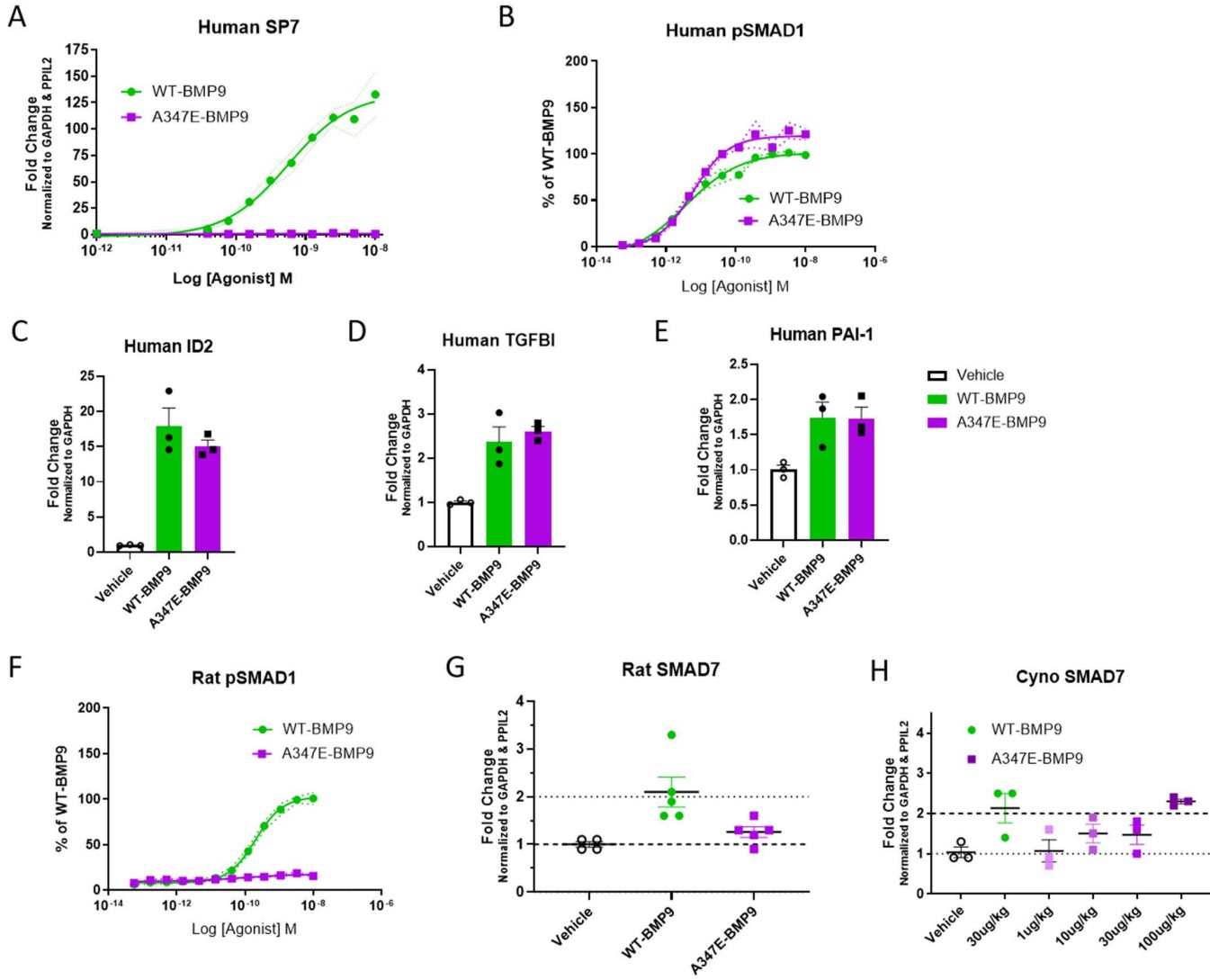

**Fig 1. Pharmacological characterization of WT BMP9 and A347E variant *in vitro* and *in vivo*. (A)** WT BMP9 and A347E variant in cell-based human MSC assay measuring SP7 expression as a marker for osteogenic differentiation in a dose-response experiment (0.0195nM-10nM). (B) pSMAD1 activation in human endothelial cells in a dose-response experiment (0.000169nM-10nM). **(C-E)** Human primary pulmonary artery endothelial cells were treated with WT BMP9 and A347E variant at 10nM for 24h and gene expression was measured using Taqman PCR **(C)** ID2, **(D)** TGFBI, **(E)** PAI1. (F) pSMAD1 activation in primary rat endothelial cells in a dose-response experiment (0.000169nM-10nM). **(G)** WT BMP9 and A347E variant (30 µg/kg) were dosed SC to naïve rats (n = 4-5). Lung tissue was harvested 6h post dose and SMAD7 expression was evaluated in lung tissue as a PD marker by Taqman PCR. **(H)** WT BMP9 (30 µg/kg; IV) and A347E variant (1, 10, 30, 100 µg/kg; SC) was dosed to cynos (n = 3). Lung tissue was harvested 6h post dose and SMAD7 expression was evaluated in lung tissue as a PD marker by Taqman PCR.

To further evaluate the functional activity of WT BMP9 and of A347E variant *in vivo*, we tested WT BMP9 in a cyno PK/PD study which revealed a WT BMP9 plasma half-life of ~1h and elevated levels of lung tissue exposure for up to 6h (S1B Fig). A dose response study using the A347E variant (1, 10, 30 and 100 µg/kg, SC) in normal cynos was performed and compared to cyno lung samples from a study dosed with a single dose of WT BMP9 (30 µg/kg, IV). Both proteins, WT BMP9 and A347E variant activated SMAD7, a downstream target of BMPR2 activation in cyno lungs at comparable doses confirming preserved SMAD activity *in vivo* (Fig 1H). WT BMP9 resulted in a 2-fold activation of SMAD7 with a single

30 μg/kg dose, whereas A347E variant showed a 2-fold upregulation of SMAD7 with a 100 μg/kg dose suggesting slight differences in the in vivo potency of WT BMP9 and A347E variant.

### *In vivo* PK evaluation of WT BMP9 in naïve rats

Long et al. [14] reported efficacy of recombinant human BMP9 in multiple PAH rat animal models. We attempted to reproduce their data using WT BMP9 given that A347E variant is inactive in rats. The PK profile of WT BMP9 in rats following IV administration indicated a short half-life of only 4.5 min (Fig 2A), suggesting fast clearance of the protein from the circulation. The PK assay allowed for differentiation of human BMP9 from endogenous rat and cyno BMP9 proteins. Lung and kidney WT BMP9 concentrations were evaluated 1h post IV bolus administration showing a dose-dependent increase in WT BMP9 levels in the lung (0.2–0.8ng/mg). In contrast, WT BMP9 was detected in kidney at low levels (~0.1ng/mg) up to the 30 μg/kg WT BMP9 dose and levels increased significantly (~0.7ng/mg) at the 100 μg/kg WT BMP9 dose (Fig 2B). These results suggest that WT BMP9 could be retained in the lung for prolonged periods of time at low concentrations, while lung saturation may lead to a spill over into other tissues, e.g., kidney.

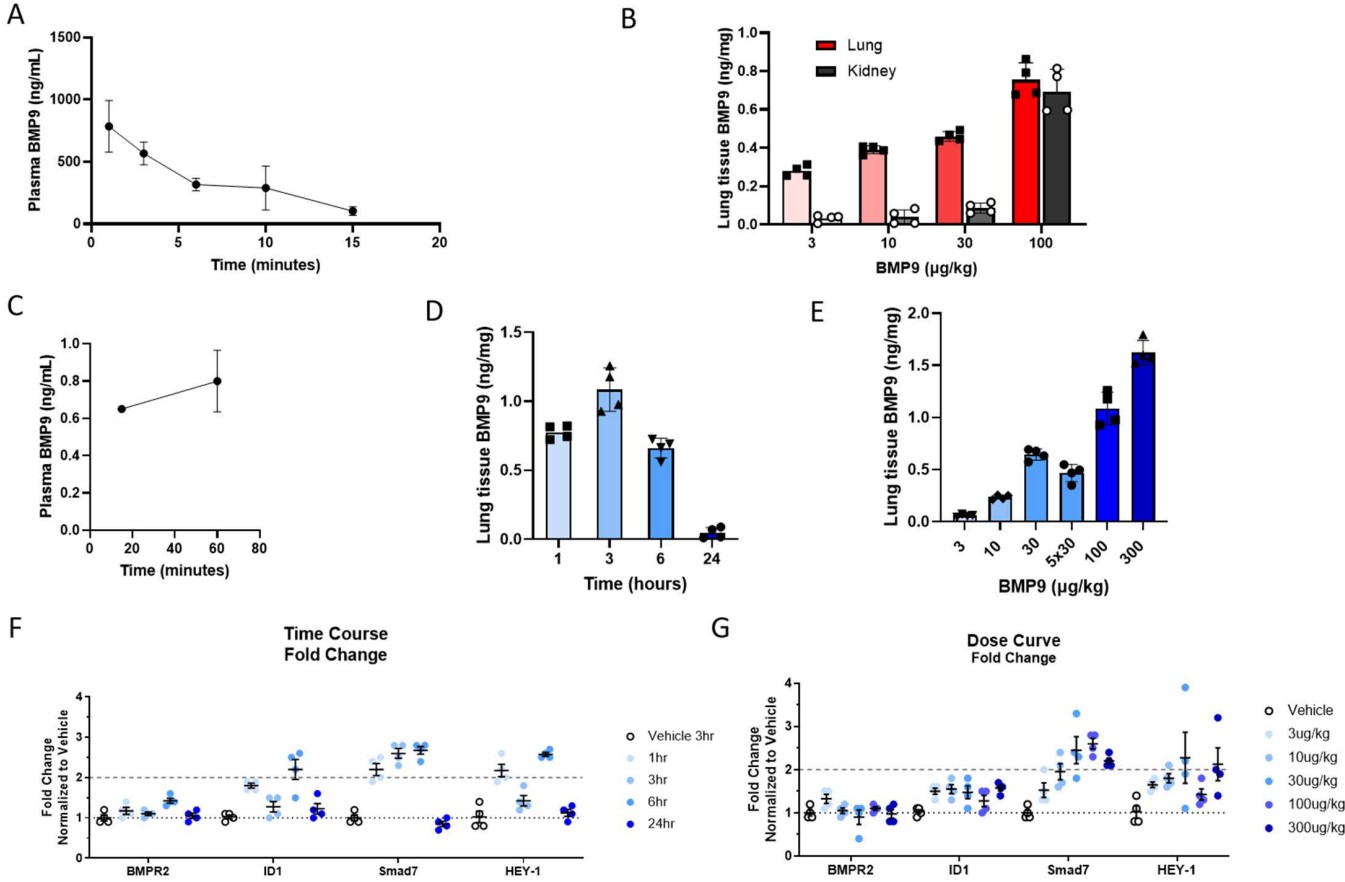

**Fig 2. *In vivo* PK evaluation of WT BMP9 in naïve SD rats. (A)** WT BMP9 (100 μg/kg; IV) plasma PK evaluation in naïve rats (n = 4). **(B)** Rat lung and kidney tissue PK (n = 4) at the indicated dose levels via IV administration 1h after dose. **(C)** WT BMP9 (100 μg/kg) rat plasma PK after SC administration (20 min: n = 1; 60 min: n = 4). **(D)** Rat lung tissue PK time course of WT BMP9 (100 μg/kg) at the indicated timepoints after a single SC administration (n = 4). **(E)** Rat lung tissue PK (n = 4) at the indicated dose levels after a single SC administration or daily injections over a 5-day period (5x30μg/kg) 3h after the last dose. **(F)** Gene expression analysis by Taqman PCR of lung tissue from PK time course study (100 μg/kg; n = 4). **(G)** Gene expression analysis by Taqman PCR of lung tissue from PK dose response study with SC administration 3h after dose (n = 4).

To better understand the PK dynamics of WT BMP9, a time course was established with single dose injections. Post SC injection (100 µg/kg WT BMP9), the plasma concentrations of WT BMP9 remained comparable (0.6–0.8ng/mg) after 20- and 60-minutes post dose at low concentrations (Fig 2C) and undetectable at later timepoints. However, in rat lung tissue elevated WT BMP9 concentrations were detected for up to 6h after a single injection, with a peak in lung tissue concentration at 3h post-dosing (Fig 2D). A dose-dependent increase in lung tissue WT BMP9 concentrations was detected at 3h after dose (Fig 2E). The 30 µg/kg dose of WT BMP9 was evaluated in a single dose experiment as well as with 5 doses on consecutive days which showed the same tissue BMP9 concentrations (Fig 2E) indicating a lack of accumulation in the lung upon daily dosing.

Target engagement (TE) of WT BMP9 was tested in naïve rats by evaluating changes in SMAD7 gene expression by qPCR. The results showed time-dependent increases in SMAD7 mRNA levels peaking at 6h post-dose and normalizing at the 24h time point (Fig 2F). Dose-response experiments showed a peak in SMAD7 fold change over baseline with 30 and 100 µg/kg, whereas higher doses did not lead to further increases in SMAD7 expression (Fig 2G), indicating saturation of the response. Additional TE markers were evaluated as well, e.g., BMPR2, ID1 and Hey-1 gene expression, but fold changes were not significant or showed a variable response to WT BMP9 stimulation. This data is suggestive of a relationship whereby WT BMP9 levels correlated with SMAD7 transcriptional induction.

## Evaluation of chronic *in vivo* efficacy in 21-day rat MCT model in prevention mode

The MCT rat model of PAH was used to evaluate the chronic efficacy of once daily WT BMP9 administration at multiple dose levels (3–100 µg/kg, SC) delivered in a disease prevention study initiating treatment at the same day as MCT dosing. The once daily dose of WT BMP9 was selected based on repeat dose experimental data confirming 24h coverage after 5 daily administrations ( Fig 4E). Imatinib (100 mg/kg, orally, once daily) was used as a positive control to reduce RVSP and RVH. Induction of PAH with MCT resulted in a reduction in body weight compared to saline-treated control rats. This effect normalized in imatinib-treated animals but not in any of the WT BMP9-treated groups (S2A Fig).

Hemodynamic parameters were measured at 21 days post-MCT induction and no changes in HR were observed in any group (S2B Fig). Heart weight to body weight (HW/BW) ratio was significantly increased (p = 0.0001) in the MCT-only group compared to no-MCT control rats (S2C Fig), and a small, but significant reduction (p = 0.0153) in the 100 µg/kg BMP9 group was observed. The Fulton index (right ventricular weight/septum + left ventricular weights (RV/LV + S)) was significantly increased (p < 0.0001) in the MCT-treated animals confirming successful induction of a PAH like phenotype (Fig 3A). WT BMP9 treatment reduced the Fulton index at higher doses, however only the 30 µg/kg dose group effect was significant (p = 0.0315). A typically observed reduction (p < 0.0001) in RV/LV + S to near control levels was observed with Imatinib treatment (Fig 3A). The RVSP was increased (p < 0.0001) in MCT treated animals and significantly reduced (p = 0.0017) only in the Imatinib group (Fig 3B). High model variability contributes to the inconsistent dose-response with WT BMP9.

Gene expression analysis of BMPR2-regulated transcripts revealed an upregulation of SMAD7 mRNA and other genes in all WT BMP9 dose groups with the maximum response at 100 µg/kg suggesting successful activation of the BMPR2/SMAD1/5/8 signaling pathway (Fig 3C). Although the rat MCT model performed as expected, as evident by the control and imatinib responses, there were no physiologically meaningful changes with WT BMP9 to improve the MCT PAH phenotype despite clear TE at 100 µg/kg with WT BMP9.

## Safety pharmacology of WT BMP9 in rats

To evaluate whether higher doses of WT BMP9 are tolerated to test the efficacy of BMP9 at higher tissue concentrations we assessed the tolerability and safety of WT BMP9 in naïve SD rats using continuous intravenous infusions at 100, 500 or 1500 µg/kg/day for 5 days. The experiment was terminated on day 4 due to lethality observed in the two high-dose groups, where 80% of rats died prior to day 4 of dosing (S3A Fig).

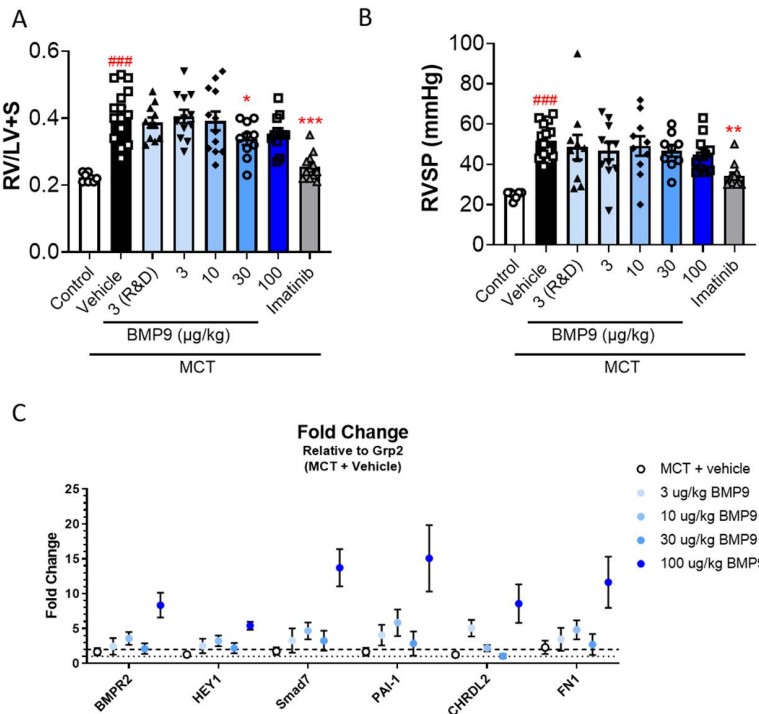

**Fig 3. Evaluation of chronic *in vivo* efficacy in 21 days rat MCT model in prevention mode.** Rats were treated with MCT (60 mg/kg; n = 15; SC), or vehicle (control; n = 8; SC) followed by daily dosing of vehicle, WT BMP9 (3-100 µg/kg; n = 12; SC) or Imatinib (100 mg/kg; n = 12; PO) for 21 days. WT BMP9 was purchased from R&D systems or manufactured in-house in *E. coli*. **(A)** Fulton index (RV/LV + S) a measure of RVH 21 days post MCT injection. Comparison vs. control: ### p < 0.0001; vs MCT + Vehicle: * p = 0.0315; *** p < 0.0001. **(B)** Invasive measurements of RVSP 21 days post MCT injection. Comparison vs. control: ### p < 0.0001; vs MCT + Vehicle: ** p = 0.0017. **(C)** Fold change vs MCT vehicle in expression of various BMPR2 target genes by Taqman PCR from lung tissue of vehicle and WT BMP9 treated groups (Error represented as standard error of the mean).

To examine the cause of the observed toxicities in a relevant PAH disease background, SD rats were implanted with telemetry devices to continuously monitor hemodynamic parameters. Rats were then treated with Su/Hx to induce a PAH like phenotype. Unexpectedly, a significant dose-dependent increase in SBP (Fig 4A) and reduction in HR (Fig 4B) were observed after a single SC dose of WT BMP9. No dose responsive changes were observed on pulmonary hemodynamics as measure by systolic pulmonary arterial pressure, only the 100 µg/kg BMP9 dose group showed changes in pulmonary hemodynamics (S4A Fig). The changes on SBP and HR relative to baseline were observed at low doses ≥10 µg/kg and were long-lasting and normalized only after 48h-dose for the 100 µg/kg dose. Peak changes in SBP (Fig 4C) and in HR (Fig 4D) were observed at around 6h. To replicate these acute effects, repeat dosing on five consecutive days of 100 µg/kg BMP9 was conducted and confirmed the presence of increased SBP (Fig 4E) and reduced HR (Fig 4F) without showing any signs of tachyphylaxis.

A study comparing daily IV (≤300 µg/kg) and SC (≤500 µg/kg) bolus injections of BMP9 did not result in lethality, but a significant reduction in body weight over 4 days of dosing in the 300 µg/kg/day IV group (Fig 4G), suggestive of animals being in distress. A trend towards dose-dependent increases in BMPR2 downstream gene expression were observed in IV and SC dosing groups with increased levels of SMAD7 and BMPR2 gene expression (Fig 4H).

In conclusion, clear PK and TE results in pre-clinical animal models, in conjunction with rat toxicology data showing adverse safety findings of human WT BMP9 and the lack of efficacy in the standard MCT rat model, a therapeutic window for treatment of PAH could not be identified.

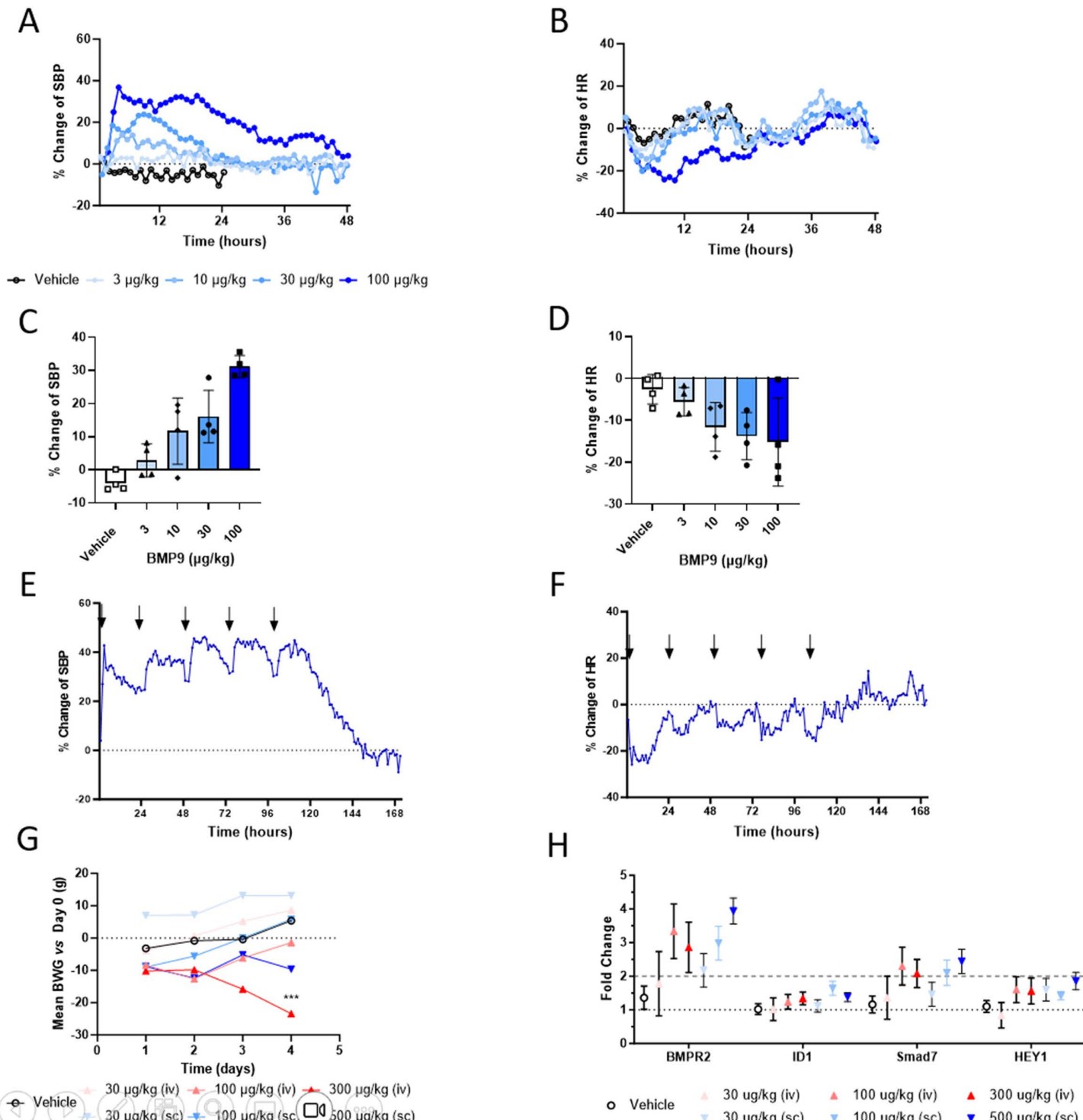

**Fig 4. Hemodynamic effects of WT BMP9 in rats. (A)** Continuous hemodynamic measurement of SBP over 48h after a single SC dose of WT BMP9 (3-100 µg/kg) in Su/Hx rats (n=6). **(B)** HR changes quantified at 6h timepoint post dose in Su/Hx rats (n=6). **(C)** SBP changes after repeat dose experiments with 5 daily doses of WT BMP9 (100 µg/kg) in Su/Hx rats (n=6). **(D)** HR measurements over 48h after a single SC dose of WT BMP9 (3-100 µg/kg) in Su/Hx rats (n=6). **(E)** SBP changes quantified at 6h timepoint post-dose in Su/Hx rats (n=6). **(F)** HR changes after repeat dose experiments with 5 daily doses of WT BMP9 (100 µg/kg) in Su/Hx rats (n=6). **(G)** Daily body weight changes during 5-day repeat dose study with a single daily dose of

WT BMP9 (30-500 μg/kg) via SC or IV routes in naïve rats (n = 5). Two-way ANOVA test with multiple comparison. Comparison vs. day 1: * p = 0.0363.
**(H)** Gene expression analysis by Taqman PCR from lung tissue at day 5 post dose in naïve rats (n = 5).

## Discussion

In this study, we identified the A347E variant as a non-osteogenic version of BMP9 with preserved SMAD1/5/8 activity in human endothelial cells and lack of SP7 transcript upregulation (osteogenic marker) in human progenitor cells *in vitro*. BMPR2 signaling activity via pSMAD1 and pSMAD3 of the A347E variant was similar to WT BMP9 in human cells. Furthermore, *in vivo* activation of SMAD7 transcripts in lungs of cynos dosed with WT BMP9 and A347E variant showed similar activity for both proteins. However, pSMAD1 activity in rat endothelial cells *in vitro* and lung tissue *in vivo* was significantly reduced with the A347E variant, limiting the possibility of validating the activity of the variant in a chronic PAH rat disease model.

Characterization of WT BMP9 PK parameters revealed relatively short plasma half-life, and long-lasting BMP9 concentrations in both rat and cyno lung tissues, suggestive of the lung acting as a sink for BMP9. These effects were consistent with the increase in BMPR2 downstream gene expression changes in the lung, which were maximally induced after 6h and normalized at 24h, despite the lack of plasma WT BMP9.

In order to validate published beneficial effects of recombinant human BMP9 in PAH rat models [14], the *in vivo* efficacy of WT BMP9 was tested in the rat MCT model. The model demonstrated the expected PAH phenotype comparing no-MCT to MCT-treated animals, and imatinib (positive control) significantly reduced the Fulton index and RVSP compared to vehicle-treated rats as previously reported [22]. In contrast, no efficacy in PAH parameters was observed with WT BMP9 despite a sustained dose-dependent increase in BMPR2-SMAD1/5/8 regulated gene expression, indicating engagement of the BMPR2 signaling pathway. No overt adverse reactions were observed during this study. These results are in contrast to Long et al. [14], who reported the efficacy of BMP9 treatment in MCT and Su/Hx rats at relatively low concentrations.

In our hands, WT BMP9 did not affect pulmonary BP in PAH animals. However, acute administration of WT BMP9 resulted in a dose-dependent increase in SBP and a corresponding reduction in HR for up to 48h limiting the potential therapeutic use of BMP9. Additional investigational studies demonstrated that WT BMP9 directly reduced endothelial cell barrier function which is a likely mechanism for the hemodynamic effects and edema [23].

Genetic data from familiar PAH patients and expression data from the idiopathic PAH patient population provide substantial evidence of reduced BMPR2 signaling as a driver of pathological vascular remodeling in PAH [24].

Effects of BMPR2 signaling in endothelial cells are well described, with loss of BMPR2 in endothelial cells resulting in a slight increase of RVSP in hypoxia-induced PAH [25]. Furthermore, Diebold et al. reported that the hypoxia-induced development of persistent pulmonary hypertension after recovery at normoxia [26] was more overt in BMPR2 knockout mice compared to WT mice. Furthermore, loss of BMPR2 in vascular smooth muscle cells was described in mice with SMC-specific knockout of BMPR2, which resulted in limited PAH phenotype after hypoxia treatment but persistent pulmonary hypertension following recovery from hypoxia, suggestive of sustained muscularization of pulmonary arteries [27].

BMP9 acts as a regulator of angiogenesis by binding and activating BMPR2/ALK1 receptor complexes, mediating anti-proliferative signals and limiting endothelial cell migration [12, 28]. The beneficial effects of BMP9 ligand therapy were postulated due to pre-clinical models of PAH and in models of lung diseases associated with pulmonary fibrosis [9, 14]. In both cases, BMP9 supplementation limits the loss of vascular endothelium and reverses vascular remodeling. Findings of hypertension were observed with overexpression of BMP10 in endothelial cell. Overexpression of BMP10, a close relative to BMP9, resulted in an increase in BP via ALK1/SMAD1 stimulation. Vice versa, the loss of BMP9 and 10 in vascular smooth muscle cells resulted in vasodilation and loss of contractile VSMC phenotype [29].

Reports from the Bailly- and Guignabert laboratories add to a more complicated picture of BMP9 as a vascular quiescence factor, whereby they demonstrated that knockout of BMP9 or inhibition of BMP9 with neutralizing antibodies

protected from PAH development in rodent models possibly due to reduced levels of Endothelin-1, a potent vasoconstrictor and contributor to vascular remodeling in PAH [30]. Recently, Desroches-Castan et al. described spontaneous arteriovenous shunts or arteriovenous malformation formations in BMP9 knockout animals, further supporting the role of BMP9 as a vascular senescence factor [31]. Mechanistically, higher VEGF/VEGFR signaling was observed in BMP9 deficient rats resulting in vasodilation, increased vessel volume in the lung, and reduced susceptibility to experimentally (chronic Hx, MCT, Su/Hx) induced PAH [8]. Similarly, in a chicken chorioallantoic membrane (CAM) assay, BMP9 administration resulted in a dose-dependent impairment of CAM angiogenesis [16], suggesting BMP9 is an angiogenesis inhibitor in vivo and acting as a quiescence factor for endothelial cells.

In summary, we could not verify the beneficial effects of BMP9 supplementation in the MCT disease model of PAH. Furthermore, BMP9 was associated with significant adverse effects on the endothelial barrier function, leading to fluid extravasation and systemic hypertension [23]. The exact molecular mechanism leading to these effects is unclear and requires further investigation. However, inhibiting BMP9 to limit its impact on pulmonary hypertension has shown beneficial effects in pre-clinical animal models of PAH [30] and Paul Yu's lab recently described in an abstract that BMP9 contributes to experimental PH by regulating expression of EC-derived molecules modulating PASMC growth and phenotype [32]. These pre-clinical results are currently being evaluated in phase 1 clinical trials utilizing a BMP9 neutralizing antibody (NCT06137742). This approach could limit the promiscuity of BMP9 on other receptor complexes and thereby provide functional improvements for PAH patients.

In conclusion, the potential beneficial effects of BMP9 ligand therapy for treating PAH are likely limited and overshadowed by the potent adverse effects that we identified and further understanding of BMP9s' mechanisms of action in regulating angiogenesis is required.

## Supporting information

**S1 Fig. Pharmacological characterization of WT BMP9 and A347E variant *in vitro* and *in vivo*.** (A) pSMAD3 activation in human endothelial cells in a dose response experiment (0.000169nM-10nM). (B) WT BMP9 (100 µg/kg; IV) plasma- and lung tissue PK evaluation in naïve cynos (n = 3).
(TIF)

**S2 Fig. Evaluation of chronic *in vivo* efficacy in 21 days rat MCT model in prevention mode.** (A) Body weight analysis of vehicle, MCT, WT BMP9 and imatinib rat groups after MCT or vehicle (control) administration (n ≥ 15/group). (B) HR (n ≥ 15/group) and (C) heart weight to body weight (HW/BW) ratio of rats 21 days post MCT induction (n ≥ 15/group). Comparison vs. control: ### $p < 0.0049$; vs MCT + Vehicle: * $p = 0.0183$.
(TIF)

**S3 Fig. Characterization of safety pharmacology of WT BMP9 in rats.** (A) Kaplan-Meier survival curve of naïve rats (n = 5) with continuous IV infusion of WT BMP9 at the indicated dose levels.
(TIF)

**S4 Fig. Characterization of systolic pulmonary arterial pressure (sPAP) of WT BMP9 in rats.** (A) Continuous hemodynamic measurement of sPAP over 48h after a single SC dose of WT BMP9 (3–100 µg/kg) in Su/Hx rats (n = 6).
(TIF)

## Acknowledgments

We thank Mark Erion, Julien Haessler, Robert Davidson, Sanjeev Bhardvaj, Chao Han, and all other team members for their support of this work.

## Author contributions

**Conceptualization:** Tobias G. Schips, Lai-ming Yung, Salam Ibrahim, Zhigang Hong, Xinkang Wang, Andrea R. Nawrocki, Jinquan Luo, Iman Farasat, Brian Geist, Yang Wang, Russell Bialecki.

**Data curation:** Tobias G. Schips, Karl W. Kavalkovich, Lai-ming Yung, Salam Ibrahim, Makhosi Edmondson, Zhigang Hong, Chin-hu Huang, Xinkang Wang, Andrea R. Nawrocki, Annmarie Winkis, Iman Farasat, Yang Wang, Jey R. Jeyaseelan, David R. Bauman.

**Formal analysis:** Tobias G. Schips, Karl W. Kavalkovich, Lai-ming Yung, Salam Ibrahim, Zhigang Hong, Chin-hu Huang, Xinkang Wang, Annmarie Winkis, Iman Farasat, Yang Wang, Jey R. Jeyaseelan.

**Funding acquisition:** Andrea R. Nawrocki, Brian Geist, David R. Bauman.

**Investigation:** Jinquan Luo.

**Methodology:** Karl W. Kavalkovich.

**Project administration:** Simon Hinke, Jey R. Jeyaseelan, David R. Bauman.

**Supervision:** Andrea R. Nawrocki, Brian Geist, Russell Bialecki, David R. Bauman.

**Visualization:** Tobias G. Schips, Karl W. Kavalkovich, Lai-ming Yung, Salam Ibrahim, Makhosi Edmondson, Zhigang Hong, Chin-hu Huang, Xinkang Wang, Simon Hinke, Annmarie Winkis, Jinquan Luo, Iman Farasat, Yang Wang, Jey R. Jeyaseelan.

**Writing – original draft:** Tobias G. Schips, Karl W. Kavalkovich, Lai-ming Yung, Salam Ibrahim, Makhosi Edmondson, Chin-hu Huang, David R. Bauman.

**Writing – review & editing:** Tobias G. Schips, Karl W. Kavalkovich, Lai-ming Yung, Makhosi Edmondson, Zhigang Hong, Chin-hu Huang, Xinkang Wang, Simon Hinke, Andrea R. Nawrocki, Annmarie Winkis, Jinquan Luo, Iman Farasat, Brian Geist, Yang Wang, Russell Bialecki, Jey R. Jeyaseelan, David R. Bauman.

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
