## [Decision Letter · Decision Letter 0]

Dear Dr. Schips,

Thank you for submitting your manuscript to PLOS ONE. After careful consideration, we feel that it has merit but does not fully meet PLOS ONE’s publication criteria as it currently stands. Therefore, we invite you to submit a revised version of the manuscript that addresses the points raised during the review process.

**In particular, you are asked to respond to the concerns regarding the results with wildtype vs. mutated BMP9, the title and the organization of the manuscript.**

We look forward to receiving your revised manuscript.

Kind regards,

Laszlo Farkas, MD

Academic Editor

PLOS ONE

**Journal Requirements:**

1. When submitting your revision, we need you to address these additional requirements. Please ensure that your manuscript meets PLOS ONE's style requirements, including those for file naming. The PLOS ONE style templates can be found at https://journals.plos.org/plosone/s/file?id=wjVg/PLOSOne_formatting_sample_main_body.pdf and https://journals.plos.org/plosone/s/file?id=ba62/PLOSOne_formatting_sample_title_authors_affiliations.pdf 2. Please update your submission to use the PLOS LaTeX template. The template and more information on our requirements for LaTeX submissions can be found at http://journals.plos.org/plosone/s/latex. 3. Thank you for stating the following financial disclosure: All authors are or were employees of Johnson and Johnson Pharmaceutical Research and Development: Janssen Research and Development LLC.This study was funded by Johnson and Johnson Pharmaceutical Research and Development: Janssen Research and Development LLC  Please state what role the funders took in the study.  If the funders had no role, please state: "The funders had no role in study design, data collection and analysis, decision to publish, or preparation of the manuscript." If this statement is not correct you must amend it as needed. Please include this amended Role of Funder statement in your cover letter; we will change the online submission form on your behalf. 4. Thank you for stating the following in the Competing Interests section: All authors are or were employees of Johnson and Johnson Pharmaceutical Research and Development: Janssen Research and Development LLC.This study was funded by Johnson and Johnson Pharmaceutical Research and Development: Janssen Research and Development LLC We note that one or more of the authors are employed by a commercial company.  a. Please provide an amended Funding Statement declaring this commercial affiliation, as well as a statement regarding the Role of Funders in your study. If the funding organization did not play a role in the study design, data collection and analysis, decision to publish, or preparation of the manuscript and only provided financial support in the form of authors' salaries and/or research materials, please review your statements relating to the author contributions, and ensure you have specifically and accurately indicated the role(s) that these authors had in your study. You can update author roles in the Author Contributions section of the online submission form. Please also include the following statement within your amended Funding Statement. “The funder provided support in the form of salaries for authors, but did not have any additional role in the study design, data collection and analysis, decision to publish, or preparation of the manuscript. The specific roles of these authors are articulated in the ‘author contributions’ section.”If your commercial affiliation did play a role in your study, please state and explain this role within your updated Funding Statement.  b. Please also provide an updated Competing Interests Statement declaring this commercial affiliation along with any other relevant declarations relating to employment, consultancy, patents, products in development, or marketed products, etc.   Within your Competing Interests Statement, please confirm that this commercial affiliation does not alter your adherence to all PLOS ONE policies on sharing data and materials by including the following statement: "This does not alter our adherence to  PLOS ONE policies on sharing data and materials.” (as detailed online in our guide for authors http://journals.plos.org/plosone/s/competing-interests) . If this adherence statement is not accurate and  there are restrictions on sharing of data and/or materials, please state these. Please note that we cannot proceed with consideration of your article until this information has been declared. Please include both an updated Funding Statement and Competing Interests Statement in your cover letter. We will change the online submission form on your behalf. 5. We note that you have included the phrase “data not shown” in your manuscript. Unfortunately, this does not meet our data sharing requirements. PLOS does not permit references to inaccessible data. We require that authors provide all relevant data within the paper, Supporting Information files, or in an acceptable, public repository. Please add a citation to support this phrase or upload the data that corresponds with these findings to a stable repository (such as Figshare or Dryad) and provide and URLs, DOIs, or accession numbers that may be used to access these data. Or, if the data are not a core part of the research being presented in your study, we ask that you remove the phrase that refers to these data. 6. Please amend either the abstract on the online submission form (via Edit Submission) or the abstract in the manuscript so that they are identical.

Reviewers' comments:

Reviewer's Responses to Questions

**Comments to the Author**

1. Is the manuscript technically sound, and do the data support the conclusions?

Reviewer #1: Yes

Reviewer #2: Yes

2. Has the statistical analysis been performed appropriately and rigorously?

Reviewer #1: Yes

Reviewer #2: Yes

3. Have the authors made all data underlying the findings in their manuscript fully available?

Reviewer #1: Yes

Reviewer #2: Yes

4. Is the manuscript presented in an intelligible fashion and written in standard English?

Reviewer #1: Yes

Reviewer #2: Yes

**Reviewer #1: ** Schips et al. studied the characterization of wild type BMP9 and a non-osteogenic variant as a potential therapeutic for pulmonary arterial hypertension. Authors concluded that the potential beneficial effects of BMP9 ligand therapy are likely limited. The findings from this study would contribute to elucidate the pathogenesis of PAH and develop the treatment for PAH. This study is interested; however, some concerns are included.

1. Authors might change the appropriate title to the results from this study.

2. Why did the WT BMP9 used this study fail to improve the PH although previous study recombinant human WT BMP9 reversed experimental PH?

3. Did authors examine the mechanism of increase BP by WT BMP9?

**Reviewer #2:**  Interesting paper

Some issues

the abstract is confusing. It should be reworded as a classical clinical abstract with intro, methods and results with a division betwenn in invo and in vitro results (usually a bit difficult for clinical reader)

Introduction should be shortened to 2 pages

Methods parts is well written, but maybe it is too long. probably a part may be put in an appendix

results:legends of figures should not be embedded

results: p values should be added

**Do you want your identity to be public for this peer review?** For information about this choice, including consent withdrawal, please see our Privacy Policy

Reviewer #1: No

Reviewer #2: **Yes: ** Fabrizio D'Ascenzo

---

## [Author Response · Author response to Decision Letter 1]

23 Jun 2025

Journal Requirements:

The revised manuscript has been updated following the PlOS One requirements.

2. Please update your submission to use the PLOS LaTeX template.

The revised manuscript has been updated following the PloS LaTeX requirements.

The new cover letter to the revised manuscript has been updated according to the instructions of the financial disclosure section

The new cover letter to the revised manuscript has been updated according to the instructions of the Competing Interests section

5. We note that you have included the phrase “data not shown” in your manuscript.

The revised manuscript does not contain the phrase “data not shown”. Three sections have been updated and references to the source data have been included. In addition new S4 Fig was added to provide source data for the lack of consistent changes in pulmonary hemodynamics with BMP9 treatment.

P21 L313

The A347E variant of human BMP9 (henceforth: A347E variant) was identified as a lead molecule based on biochemical and biophysical properties[21].

Added reference #21 and removed data not shown

P25 L409 (track changes); P24 L408 (clean)

The once daily dose of WT BMP9 was selected based on repeat dose experimental data confirming 24h coverage after 5 daily administrations (Fig 4 E).

Added reference to Fig 4E and removed data not shown

P27 L455 (track changes); P26 L454 (clean)

No dose responsive changes were observed on pulmonary hemodynamics as measure by systolic pulmonary arterial pressure, only the 100µg/kg BMP9 dose group showed changes in pulmonary hemodynamics (S4A Fig).

Added new S4A Fig to the manuscript and reference to S4A Fig and removed data not shown

6. Please amend either the abstract on the online submission form (via Edit Submission) or the abstract in the manuscript so that they are identical.

We updated the abstract in the manuscript and the online submission form, but on our end, both were already identical.

Comments to the Author

Reviewer #1: Schips et al. studied the characterization of wild type BMP9 and a non-osteogenic variant as a potential therapeutic for pulmonary arterial hypertension. Authors concluded that the potential beneficial effects of BMP9 ligand therapy are likely limited. The findings from this study would contribute to elucidate the pathogenesis of PAH and develop the treatment for PAH. This study is interested; however, some concerns are included.

1. Authors might change the appropriate title to the results from this study.

We appreciate the reviewer's suggestion to revise the manuscript title. The new title, "In vitro and in vivo Characterization of Wild Type BMP9 and a Non-Osteogenic Variant in Models of Pulmonary Arterial Hypertension," emphasizes the characterization of BMP9 and excludes its therapeutic potential in PAH.

2. Why did the WT BMP9 used this study fail to improve the PH although previous study recombinant human WT BMP9 reversed experimental PH?

We appreciate the reviewer's question regarding our inability to reproduce previously published results demonstrating that WT BMP9 could reverse experimental PAH. The reasons for the conflicting findings on the effects of WT BMP9 in rat PAH models remain unclear. Our team explored several options to identify the root cause of the lack of efficacy observed.

We evaluated a commercial version of WT BMP9 from R&D Systems and an internally produced WT BMP9 protein from E. coli. Both proteins displayed indistinguishable activity in vitro, specifically in pSMAD1 activation in human endothelial cells. Various formulations (PBS with or without serum albumin) were tested for solubilizing the proteins, but no differences in WT BMP9 activity were noted. Additionally, we conducted multiple in vivo studies to assess the pharmacokinetic and pharmacodynamic parameters of WT BMP9 in rats and established a PK/PD relationship by examining its effects on upregulating transcriptional targets such as SMAD7 (Fig 2).

In long-term efficacy studies using the MCT rat model of PAH, we observed activation of downstream transcriptional mediators (Fig 3C), but no consistent improvements in pulmonary function. The MCT model worked as expected in the vehicle control group (significant increases in RVSP and Fulton index) as well as the positive control group receiving PDGFR inhibitor Imatinib (significant decreases in RVSP and Fulton index). This outcome was repeated at least three times with similar results. Furthermore, we assessed treatment paradigms reported by Long et al. (2016, Nature Medicine) in the Sugen/Hypoxia rat model of PAH, yet found no efficacy of WT BMP9 in ameliorating PAH pathology. Long et al. reported using 600 ng/day BMP9 via intraperitoneal injections (~0.15 μg/kg). We tested various injection routes and found the subcutaneous route to be the most reliable and efficacious for dosing WT BMP9 in rodents. In our hands, the minimum effective dose that consistently activated the BMPR2 pathway was 3 μg/kg, approximately 20 times higher than the dose used by Long et al. Since no study evaluating BMP9 in models of PAH has evaluated pharmacokinetic and pharmacodynamic parameters of BMP9 proteins in rats, it is difficult to directly compare different studies.

In summary, our evaluation of WT BMP9's effects failed to reproduce its therapeutic benefits in rat PAH models. Toxicology study results suggest that WT BMP9 treatment induces an immediate increase in systemic blood pressure, which may interfere with measurements of pulmonary hemodynamic improvements.

Furthermore, published literature from Tu et al. (Circ Res 2019 Mar 15;124(6):846-855; doi: 10.1161/CIRCRESAHA.118.313356; Selective BMP-9 Inhibition Partially Protects Against Experimental Pulmonary Hypertension) suggests a more complex role for BMP9 in the development of PAH than shown in the study of Long et al. (Nat Med 21, 777–785 (2015). doi: 10.1038/nm.3877; Selective enhancement of endothelial BMPR-II with BMP9 reverses pulmonary arterial hypertension).

In addition, Yong et al. (Circulation 2023 Nov 6;Vol 148, Suppl_1; doi: 10.1161/circ.148.suppl_1.18578; Abstract 18578: BMP9 Regulates Endothelial Expression of Mediators of Pulmonary Vascular Remodeling) presented an abstract suggesting that recombinant BMP9, BMP9/BMP10 trap ALK1-Fc, and BMP9 neutralizing antibody can prevent or reverse experimental PH when administered before or after onset of experimental PH. However, recombinant disulfide-linked BMP9 was not protective in SUGEN/Hypoxia (SU-Hx)- or monocrotaline (MCT)-induced experimental PH, suggesting dissociated monomers might function as competitive BMP9 antagonists. These statements provide further evidence that BMP9 supplementation therapy might not be a beneficial treatment option for patients suffering from PAH.

We included the Yong et al. citation in the revised discussion which now provides further evidence and explanation for our study results.

3. Did authors examine the mechanism of increase BP by WT BMP9?

We appreciate the reviewer's question regarding how WT BMP9 raises blood pressure in rats. Our studies identified vascular leakage in major organs as the root cause, leading to distress in low-dose groups and lethality in high-dose groups. These findings were presented at a conference (Kanerva, 2024), and we are preparing an additional manuscript for publication.

Reviewer #2: Interesting paper

Some issues

1. the abstract is confusing. It should be reworded as a classical clinical abstract with intro, methods and results with a division betwenn in invo and in vitro results (usually a bit difficult for clinical reader)

We appreciate the reviewer's suggestion to change the abstract format for better clarity. We have revised it to separate the introduction, methods, and results, further dividing the results into in vitro and in vivo findings.

2. Introduction should be shortened to 2 pages

We acknowledge the reviewer's request to shorten the introduction. Due to the extensive published information on this topic, it is challenging to condense the introduction to two pages. After consulting with the Editor, we have decided that the introduction adequately reviews the topic and will remain in its current form.

3. Methods parts is well written, but maybe it is too long. probably a part may be put in an appendix

We appreciate the reviewer's request to change the format of the method section. After consulting with the Editor, it has been decided that the method section provides a comprehensive summary of the methods used and will remain in its current form.

4. results:legends of figures should not be embedded

We thank the reviewer for the request to change the format of the result section. PLOS formatting requirements involve the figure legends to be embedded in the manuscript text after first mention in the corresponding section and after consulting with the Editor we decided to keep the current format of the figure legends.

5. results: p values should be added

We appreciate the reviewer's request to add p-values. The p-values were presented in the figure legends, and we agree that including them in the results section text would enhance the assessment of the results. Therefore, we have added p-values to the results section.

P24 L414/415:

The heart weight to body weight (HW/BW) ratio was significantly increased (p=0.0001) in the MCT-only group compared to no-MCT control rats (S2C Fig). Additionally, a small but significant reduction (p=0.0153) was observed in the 100µg/kg BMP9 group.

P25 L417/420/421/422:

The Fulton index (right ventricular weight/septum + left ventricular weights (RV/LV+S)) was significantly increased (p<0.0001) in the MCT-treated animals, confirming the successful induction of a PAH-like phenotype (Fig 3A). WT BMP9 treatment reduced the Fulton index at higher doses; however, only the effect of the 30µg/kg dose group was significant (p=0.0315). A typical reduction (p<0.0001) to near control levels was observed with Imatinib treatment (Fig 3A). The RVSP was increased (p<0.0001) in MCT-treated animals and significantly reduced (p=0.0017) only in the Imatinib group (Fig 3B).

---

## [Decision Letter · Decision Letter 1]

In vitro and in vivo characterization of wild type BMP9 and a non-osteogenic variant in models of pulmonary arterial hypertension

PONE-D-25-06191R1

Dear Dr. Schips,

We’re pleased to inform you that your manuscript has been judged scientifically suitable for publication and will be formally accepted for publication once it meets all outstanding technical requirements.

Kind regards,

Laszlo Farkas, MD

Academic Editor

PLOS ONE

Additional Editor Comments (optional):

Reviewers' comments:

Reviewer's Responses to Questions

**Comments to the Author**

Reviewer #1: All comments have been addressed

Reviewer #2: All comments have been addressed

2. Is the manuscript technically sound, and do the data support the conclusions?

Reviewer #1: Yes

Reviewer #2: Yes

3. Has the statistical analysis been performed appropriately and rigorously?

Reviewer #1: Yes

Reviewer #2: Yes

4. Have the authors made all data underlying the findings in their manuscript fully available?

Reviewer #1: Yes

Reviewer #2: Yes

5. Is the manuscript presented in an intelligible fashion and written in standard English?

Reviewer #1: Yes

Reviewer #2: Yes

Reviewer #1: Authors changed the title and addressed the reviewer's comments. This study do not have concerns about reserach ethics or publication ethics.

Reviewer #2: All comments have been addressed and authors should be complimented for. No other comments and improvement is needed

**Do you want your identity to be public for this peer review?** For information about this choice, including consent withdrawal, please see our Privacy Policy

Reviewer #1: No

Reviewer #2: **Yes: ** Fabrizio D'Ascenzo

---

## [Editor Report · Acceptance letter]

PONE-D-25-06191R1

PLOS ONE

Dear Dr. Schips,

I'm pleased to inform you that your manuscript has been deemed suitable for publication in PLOS ONE. Congratulations! Your manuscript is now being handed over to our production team.

Kind regards,

on behalf of

Dr. Laszlo Farkas

Academic Editor

PLOS ONE